

# Vinasse application and cessation of burning in sugarcane management can have positive impact on soil carbon stocks

Caio F. Zani[1,2], Arlete S. Barneze[2,3], Andy D. Robertson[2,4], Aidan M. Keith[2], Carlos E.P. Cerri[5], Niall P. McNamara[2] and Carlos C. Cerri[1,5,†]

[1] Centro de Energia Nuclear na Agricultura, Universidade de São Paulo, Piracicaba, São Paulo, Brazil
[2] Lancaster Environment Centre, Centre for Ecology & Hydrology, Lancaster, United Kingdom
[3] Soil and Ecosystem Ecology Laboratory, Lancaster Environment Centre, Lancaster University, Lancaster, United Kingdom
[4] Institute of Biological and Environmental Sciences, University of Aberdeen, Aberdeen, United Kingdom
[5] Departamento de Ciência do Solo, Escola Superior de Agricultura "Luiz de Queiroz", Universidade de São Paulo, Piracicaba, Brazil
[†] Deceased.

Corresponding author
Caio F. Zani, caiozani@usp.br, caiofzani@gmail.com

## ABSTRACT

Bioenergy crops, such as sugarcane, have the potential to mitigate greenhouse gas emissions through fossil fuel substitution. However, increased sugarcane propagation and recent management changes have raised concerns that these practices may deplete soil carbon (C) stocks, thereby limiting the net greenhouse gas benefit. In this study, we use both a measured and modelled approach to evaluate the impacts of two common sugarcane management practices on soil C sequestration potential in Brazil. We explore how transitions from conventional (mineral fertiliser/burning) to improved (vinasse application/unburned) practices influence soil C stocks in total and in physically fractionated soil down to one metre. Results suggest that vinasse application leads to an accumulation of soil C of 0.55 Mg ha$^{-1}$ yr$^{-1}$ at 0–30 cm depth and applying unburned management led to gains of ∼0.7 Mg ha$^{-1}$ yr$^{-1}$ at 30–60 cm depth. Soil C concentration in the Silt+Clay fraction of topsoil (0–20 cm) showed higher C content in unburned management but it did not differ under vinasse application. The CENTURY model was used to simulate the consequences of management changes beyond the temporal extent of the measurements. Simulations indicated that vinasse was not the key factor driving increases in soil C stocks but its application may be the most readily available practice to prevent the soil C losses under burned management. Furthermore, cessation of burning may increase topsoil C by 40% after ∼50 years. These are the first data comparing different sugarcane management transitions within a single area. Our findings indicate that both vinasse application and the cessation of burning can play an important role in reducing the time required for sugarcane ethanol production to reach a net C benefit (payback time).

## INTRODUCTION

Sugarcane has the potential to contribute significantly towards a renewable energy future (*Goldemberg et al., 2014*; *OECD/FAO, 2013*). Ethanol production derived from sugarcane has become of global interest due to its potential to mitigate emissions of greenhouse gases (GHGs) through fossil fuel substitution (*Goldemberg et al., 2014*). There are, however, concerns that bioenergy-driven land use change (LUC) may adversely impact soil carbon (C) stocks (*Guo & Gifford, 2002*) and increase the amount of time required before a biofuel feedstock can achieve a net positive GHG benefit when displacing fossil fuels (i.e., its *payback time*) (*Fargione et al., 2008*; *Mello et al., 2014*).

Whilst LUC to sugarcane always impacts the overall C balance, the extent of this impact is often management-specific and therefore exact payback time for any given site is directly related to how it is managed (*Davis et al., 2013*; *Walter et al., 2011*). Management practices which increase bioenergy crop yields, increase soil C accumulation and decrease GHG emissions are an important part of sustainable land use in light of climate change (*Lal, 2004*; *Smith et al., 2012*). With the expected increase in sugarcane use as a biofuel in the near future (*Goldemberg et al., 2014*) the current and expanding sugarcane areas have received considerable attention regarding two key management practices: (1) the fate of by-products from the ethanol cycle to the field (e.g., vinasse); and (2) reduced biomass removal due to a transition from traditional pre-harvest burning to an unburned management.

Vinasse is produced from the processes of producing ethanol from sugarcane; for each litre (L) of sugarcane ethanol produced around 13 L of this liquid by-product is generated. Sugarcane vinasse is a source of soluble C but also has a high organic content (chemical oxygen demand-COD, 50–150 kg m$^{-3}$) and is highly concentrated in potassium (K) ($\sim$2 kg m$^{-3}$) (*Christofoletti et al., 2013*). A moderate amount of other minerals, such as nitrogen (N), calcium (Ca), phosphorus (P) and magnesium (Mg) (0.28, 0.33, 0.10 and 0.13 kg m$^{-3}$, respectively) are also found in vinasse (*De Resende et al., 2006*). Vinasse can be applied by irrigation (ferti-irrigation) to sugarcane fields as a simple and inexpensive soil fertiliser which reduces chemical input requirements (*Laime et al., 2011*) and increases sugarcane yields (*De Resende et al., 2006*). It is estimated that over 80% of São Paulo State is under vinasse ferti-irrigation (*Fronzalia, 2007*). A recent evaluation by *Prado, Caione & Campos (2013)* also suggests that vinasse application to sugarcane fields can deliver benefits to the soil when applied at appropriate levels ($\sim$200 m$^3$ ha$^{-1}$) including an increase in soil C stocks (*De Resende et al., 2006*). By contrast, an inappropriate and/or overuse of vinasse application might cause environmental issues including soil salinisation (*Christofoletti et al., 2013*) and potential nutrient discharge into water bodies leading to eutrophication (*Có Júnior, Marques & Tasso Júnior, 2008*). Furthermore, recent studies suggest that vinasse leads to GHG emissions during its storage, transportation and application to the soil (*Oliveira et al., 2013*; *Oliveira et al., 2015*). Whilst the vast majority of studies have focused on GHG emissions and perceive vinasse discharge as a pollutant (see review by *Fuess & Garcia, 2014*) there is also a need to consider its potential benefits to soil C. To the best of our knowledge, there is only one study evaluating the impacts of vinasse application on

soil C stocks, reporting a non-significant increase of 0.25 Mg ha$^{-1}$ yr$^{-1}$ in the top 20 cm (*De Resende et al., 2006*).

The burning sugarcane residues on the soil surface prior to manual harvesting has historically been a common management practice in Brazil and worldwide. This burning process facilitates manual harvesting and subsequent sugarcane transport by removing leaf material but the cane remains intact. This management practice is now considered to be particularly detrimental both in terms of climate change and human health due to GHG and black C emissions (*Cançado et al., 2006*; *Galdos et al., 2013*). Recent studies highlight that unburned sugarcane management can enhance C sequestration in soils, with a reported increase in topsoil (30 cm) C stocks of 1.50 Mg ha$^{-1}$ yr$^{-1}$ (*Cerri et al., 2011*; *Galdos et al., 2010*). However, these previous studies did not use a paired site approach to directly examine the impacts of cessation of burning (i.e., the transition in management practices). Furthermore, most previous sugarcane studies only examine change in topsoil; comparable research under European perennial bioenergy crops has demonstrated that C in deeper soil layers must be included in any bioenergy sustainability assessment (*Keith et al., 2015*; *Rowe et al., 2016*; *Walter, Don & Flessa, 2015*). While there are undeniable benefits regarding cessation of burning such management changes can also lead to soil compaction from mechanised harvesting, decreased crop yields, and requiring more frequent soil tillage (*Silva-Olaya et al., 2013*; *Walter et al., 2014*).

Management mediated changes in soil C stocks might be underpinned by changes in the soil organic matter (SOM) C associated with different soil particle sizes and mineralogical compositions (*Christensen, 1992*). Evaluating the C content of different particle-size fractions can therefore improve our understanding of the mechanisms by which the soil C stabilisation occurs, as well as the sensitivity of SOM content in response to management practice (*Brandani et al., 2016*). Recent studies appraising different sugarcane management practices have reported changes in C measured in different soil fractions (*Brandani et al., 2016*; *Lisboa et al., 2009*; *Signor et al., 2014*). *Brandani et al. (2016)* highlighted that the cessation of burning combined with organic fertiliser inputs is able to accumulate soil C in at least three SOM fractions (<53 µm; 75–53 µm and 2,000–75 µm). Additionally, *Signor et al. (2014)* showed that cessation of burning can also lead to increased soil C stocks particularly in organo-mineral complexes that typically have a higher mean residence time in the soil than particulate organic matter. Ultimately, research into the impacts of changing management practices on soil C stocks and soil C fractions can help reduce the uncertainty of calculating payback time associated with sugarcane grown for ethanol production (such as those reported by *Mello et al. (2014)*).

Empirical research of soil C dynamics often provide us with a single snapshot in time (e.g., a paired-site approach). To accurately determine the true long-term impacts of a management change, we must also compliment these measurements with informed predictions that can interpolate between the individual, measured time-points. Mathematical models can help with this by simulating the potential influence that different management practices have on soil C stocks. One of the most well-known ecosystem models (the CENTURY model version 4.0; *Parton et al. (1987)* simulates elemental cycling through plant production and soils, and has been validated to accurately simulate annual

and perennial crops, forests and pastures (*Cerri et al., 2007*; *Cong et al., 2014*; *Parton & Rasmussen, 1994*). CENTURY has also been reported to provide reliable simulations of sugarcane plantations (*Brandani et al., 2015*; *Galdos et al., 2009a*; *Galdos et al., 2009b*; *Vallis et al., 1996*) and thus can improve our understanding in the context of management practice changes.

This study aims to address the impacts of vinasse application and the cessation of burning on soil C stocks at the same location. Using a paired-site approach with previous management practices as a reference, we explore how transitions to new management might influence soil C stocks and C fractions, down to 1 metre. Additionally, we use the process-based model, CENTURY, to evaluate if either vinasse application, burning, cessation of burning or a combination of these practices, could result in longer term changes in soil C stocks. A simplistic recalculation of the payback time for soil C stocks and sugarcane ethanol is also provided taking into consideration the management changes assessed here.

## MATERIAL AND METHODS

### Site selection and description

A paired-site (space-for-time) approach was used at two locations to quantify changes in soil C stocks after changes in management practices, with the previous and the current management at the same location. The pairings were deemed suitable since they had similar soil type, climatic conditions and a reference sugarcane management area to pair with the new management approach, a no vinasse-to-vinasse (NV-to-V) and a burned-to-unburned (B-to-UB) comparison. These sites were located in Ourinhos, São Paulo State (22°59′S, 49°52′W, 492 m above sea level) (Fig. S1). The climate of the region is classified as tropical savanna with hot humid summers and cold dry winters according to the Köppen classification (Cwa tropical). The average annual precipitation and temperature are 1,321 mm and 22.8 °C, respectively (Fig. S2).

The soil was classified as Rhodic Ferralsol (*FAO, 2014*); soil properties are shown in Tables S1 and S2. Soil under typical native vegetation (F site) in the region, used as a baseline for modelling approaches, was sampled nearby of the study sites (Lat.: 23°05′08″S; Long.: 49°37′52″W), having the same soil type as the sugarcane planted areas (more information can be found in *Cherubin et al. (2015)* and *Franco et al. (2015)*). This native vegetation was identified as a seasonal semi-deciduous forest within the Atlantic forest biome, compromising a transitional region between the Atlantic rainforest and Cerrado vegetation (*Cherubin et al., 2015*; *Franco et al., 2015*).

### Management comparisons
#### *No Vinasse (NV) to Vinasse (V)*
The NV-to-V location has been cultivated with sugarcane (38.63 ha) since 1958 after conversion from pasture. Since 2004, 12.72 ha of this area has received vinasse application (V site), whereas the remaining area has continued to receive conventional mineral fertiliser (NV site) (Fig. 1A). In 2008 and 2009 mechanised harvesting was also introduced at the V and NV sites respectively, and before that burning practice followed by manual harvesting

A) No Vinasse (NV) to Vinasse (V)

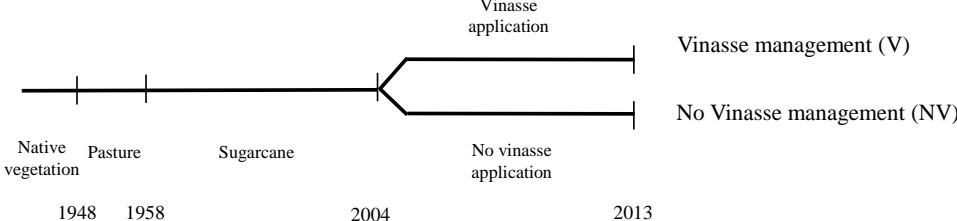

B) Burned (B) to Unburned (UB)

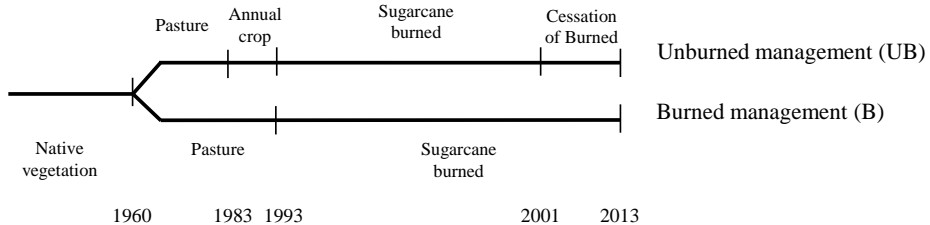

**Figure 1** **Timeline of land use and management changes at each site assessed (A) No Vinasse *to* Vinasse, (B) Burned *to* Unburned.**

was employed at both. Management included, in addition to vinasse conventional fertiliser, N application in 2013 (164 kg ha$^{-1}$ and 192 kg ha$^{-1}$ for V and NV, respectively), and P application in January 2012 (120 kg ha$^{-1}$, at both the V and NV sites) and tillage in 2011. At the NV site, potassium chloride (KCl) was applied as an inorganic fertiliser (520 kg ha$^{-1}$) at the time of sugarcane planting, whilst vinasse was applied in the V site each year since 2004 at an average rate of 268 m$^3$ ha$^{-1}$. Taking into account the K content typically found in the vinasse (from 0.84 to 1.8 g L$^{-1}$) these applications correspond to a total rate of around 440 kg ha$^{-1}$ of K concentration (*Oliveira et al., 2013*; *Oliveira et al., 2015*).

### Burned (B) to Unburned (UB)

The B-to-UB location comprised of 30.5 ha of forested land converted to pasture in 1960 (Fig. 1B). The area was converted to sugarcane in 1993 but prior to sugarcane, where the UB site is located, annual cropping was used for 10 years. From 1993 to 2001, the whole area was harvested using burning methods. From 2001 the plantation was split into two: burned practices with manual harvesting continued at one site (B site, 15.48 ha), and ceased at another, employing mechanised harvesting instead (UB site, 15.02 ha) (Fig. 1B). This particular change in the management also resulted in a straw deposition in the soil surface of roughly 10 to 20 Mg ha$^{-1}$ every year. In the year of sampling (2013), the B site had been under the burned management for 20 years, while the UB site had undergone 8 years of burned management practice followed by 12 years of unburned practices.

The B and UB sites were cultivated by conventional tillage with added mineral fertilisers. Nitrogen fertiliser was consistently applied at a rate of 86 kg ha$^{-1}$ yr$^{-1}$ from 2009. During planting, the B site received 119 kg ha$^{-1}$ of phosphorus (P) and 521 kg ha$^{-1}$ of potassium (K), whereas the UB site received 494 kg ha$^{-1}$ of K with no P addition. The last tillage of the sites occurred in 2008 and 2011 for U and UB, respectively.

## Approach, sampling method and measurements

The experimental design followed the schematic suggested by *Cerri et al. (2013)* and used by *Mello et al. (2014)* (Fig. S3), but included more detailed sampling of soil layers. Specifically, the sampling grid for each study field comprised nine trenches distributed in 3 ×3 grid spaced at 50 m apart dug in each management site and covering approximately 1 ha (100 ×100 m). In each of the three deeper trenches (Fig. S3; dimensions 120 × 120 × 120 cm) soil samples were taken in 10 cm increments to 100 cm, from the opposing walls of the trench to obtain representative soil samples between and within plant rows. Shallow trenches (Fig. S3; six trenches with dimensions 40 × 40 × 40 cm) were sampled to 30 cm depth also in 10 cm increments, only from one wall. In each layer of each trench two types of samples were taken, one for bulk density (BD), C and N assessments, using stainless steel rings with a diameter and height of 5 cm (98.17 cm$^3$ internal ring volume) (78 samples), and another for chemical and physical analyses (60 samples). All soil samples were taken in July 2013.

### Soil analyses

Soil samples taken from the 120 cm deep trenches were used to characterise the study sites in terms of texture (sand, silt and clay content), soil pH (CaCl$_2$) and the concentration of macro and micronutrients, according to the methods provided in *Anderson & Ingram (1989)* and *EMBRAPA (1997)* (Tables S1, S2). Soil texture was derived for each 10 cm increment down to 100 cm, while only certain depths (0–10; 10–20; 20–30; 40–50 and 90–100 cm) were used for the other analyses (one composite sample from the three deeper trenches).

Soil moisture content was derived for all samples by drying the samples to constant weight at 105 °C. To determine total C and N, samples were ground to a fine powder and sieved to 150 μm prior to analyses. Total soil C and N were determined by dry combustion (*Nelson & Sommers, 1996*) using a Leco Truemac CN elementary analyzer (furnace at 1,350 °C in pure oxygen).

### Physical fractionation

Physical fractionation was done according to *Christensen (1992)*, which separates the particle sizes soil by dispersion, wet sieving, flotation and sedimentation, followed by a subsequent mass balance check. The method includes separation of the uncomplexed organic matter and primary organo-mineral complexes (*Christensen, 2001*).

The physical fractionation was carried out for all studied areas for the following depths: 0–10 cm, and 10–20 cm (3 replicates), 40–50 and 90–100 cm (1 replicate). Briefly, 20 g of dry soil was sonicated in 70 mL of deionised water at 500 W for 15 min (providing roughly 13 J per sample) using an ultrasonic processor (Model VC-505; Sonics Vibra Cell). Subsequently, the sample was wet sieved through a 53 μm sieve to separate the sand

(assigned here as heavy fraction HF >53 μm) and light-coarse particulate organic matter (POM >53 μm) from silt and clay (S + C <53 μm) particles (Fig. S4). POM and HF fractions which were retained on the sieve were separated by flotation and sedimentation using deionised water (1 g cm$^{-3}$) (Fig. S4). All resulting fractions were air-dried, weighed and the sum of the fraction checked against the original 20 g. Lastly, the C content of each fraction was determined using dry combustion methods as for total C and N analyses.

### Soil C stock and payback time recalculation

Soil C stock was calculated for each 10 cm layer by relating the soil C concentration to bulk density and depth increment. As soil C stock is directly related to bulk density which can be altered by different management practices, it was necessary to adjust the mass of the soil layers being compared to a reference mass (average bulk density from previous management carried out) according to *Ellert & Bettany (1995)*. The use of C content and bulk density data were used to make the comparison of soil C stocks between the paired sites in order to derive the equivalent soil mass across each management comparison.

The payback time for the soil C stock debt and sugarcane ethanol production was recalculated following the same method utilised by *Mello et al. (2014)* but here considering the management change (no vinasse-to-vinasse and burned-to-unburned) effects. The response ratios denoted as management change factors (i.e., the proportional change in the soil C stocks due to management change) were calculated using the average soil C stocks for each management practice assessed over the layer increments 0–30, 0–50 and 0–100 cm. The management change factor higher than 1 indicates soil C stock gain while values lower than 1 represents losses. The C debt (Mg CO$_2$ ha$^{-1}$) after 20 years due to either LUC from pasture or cerrado vegetation to sugarcane plantations reported by *Mello et al. (2014)* were used to recalculated the payback time. The estimated payback time included our findings after management changes (NV-to-V and B-to-UB), plus the sugarcane ethanol offset of 9.8 Mg CO$_2$ ha$^{-1}$ yr$^{-1}$ reported by *Fargione et al. (2008)*.

## CENTURY model description and assessment

The CENTURY model version 4.0 was used to simulate long-term changes in both soil C stocks for the upper 20 cm of the soil profile and to simulate sugarcane yields. Briefly, CENTURY is a mechanistic model at ecosystem scale that operates on a monthly time step; it has interacting crop production routines and soil modules to partition SOM into three pools with different turnover times controlled by specific decomposition rates (1. active, fast turnover, 2. intermediate, medium turnover, 3. passive, slow turnover).

### Initialisation, parameterisation and assessment of the model

The model was initialised by inputting site-specific features including soil texture data, bulk density and pH from native forest vegetation (F site) nearby the sugarcane study sites (*Cherubin et al., 2015*; *Franco et al., 2015*). Meteorological inputs including mean precipitation and monthly temperatures (minimum and maximum), taken by the Instituto Nacional de Metereologia (*INMET, 2016*) from the weather station located in Presidente Prudente (170 km from the sites), were used for the period between 1961–2008. Site specific meteorological inputs were used from 2009 to 2013, taken from a weather station located

at the São Luiz mill (Ourinhos). The combination of these two stations was necessary due to limited weather data being available for the specific study site prior to 2009.

Pre-cultivation conditions were first run to an equilibrium state (7,000 years) under native vegetation. Model parameterisation was required for this native vegetation situation, since the location of native vegetation is under a transitional situation between Cerrado vegetation and Atlantic Rainforest. The parameterisation was made in accordance with data reported in *Cherubin et al. (2015)* and *Franco et al. (2015)* and used the average between default values of Cerrado and Atlantic rainforest specified by CENTURY for the remaining parameters. A fire event was scheduled every 120 years during the equilibrium spin-up to simulate forest disturbance based on the premise that there is a gap formation due to the occurrence of tree mortality and tree-fall (*Cerri et al., 2004*). Subsequent checks were performed to ensure that simulated above (leaves, branches and tree large woods) and belowground (roots) biomass was in line with measured data from published studies (*Cunha et al., 2009*; *Vieira et al., 2011*). Once the SOM reached the equilibrium state, a deforestation event was simulated in accordance with the LUC procedure parameterised by *Cerri et al. (2004)*, which involves slash and burn of the native vegetation.

Each simulation scenario was scheduled to represent the management history of each 'site', summarised in Fig. 1. Soil texture was kept constant across each study sites (66, 19 and 14% for clay, sand and silt, respectively) as the soil under native vegetation and our sampled sugarcane sites were similar. Owing the lack of information about pasture and soybean management used prior to conversion into sugarcane plantations, we assumed a typical management practice (i.e., degraded pasture), characterised by low to moderate productivity mainly caused by the lack of soil management (liming, fertilisation) and overgrazing, as well as monocropping and conventional soil tillage for soybean (plowing and disking) as reported by *Batlle-Bayer, Batjes & Bindraban (2010)* and *Maia et al. (2010)*. Simulated results of these practices were validated (above and belowground) in relation to biomass productivity stated in other studies (*Bordin et al., 2008*; *Finoto et al., 2012*; *Lilienfein & Wilcke, 2003*). The model was run with sugarcane cultivation based on parameterisation developed and reported by *Galdos et al. (2009a)*; *Galdos et al. (2009b)* (Tables S3–S5).

### Long-term scenarios

Simulations were extended beyond the 2013 measurement date to predict the effects of management practice changes up to 2100, where a near-equilibrium state of soil C stocks was achieved for all management practices. In this procedure, monthly weather conditions were established on the average historic data (1961–2013) assuming no climate change or variation in atmospheric $CO_2$ concentrations. The sugarcane crop cycle was fixed to imply a rotation with 6 years of annual harvest followed by conventional tillage and immediate replanting after the 6th harvest. Additional simulations were performed to evaluate the influence of varying application rates of vinasse, including a combination of a burned management and vinasse application at high and low application rates, 400 and 100 m$^3$ ha$^{-1}$ yr$^{-1}$ respectively.

## Statistical analyses

Differences in soil C stock depth profiles due to management practice changes were tested with a bootstrap re-sampling and Loess regression approach (*Keith et al., 2016*). Firstly, the combined land management data (i.e., NV-to-V and B-to-UB) was re-sampled by bootstrap with replacement ($n = 1,000$). These data were then modelled using Loess regression and the bootstrap samples used to generate 95% confidence intervals around a modelled soil C profile. This represented the null hypothesis that there was no difference between the land management practices. The data from the new land management practice only was then modelled using Loess regression; if the modelled line for this soil C profile sat outside the confidence interval based on the null hypothesis, it was inferred that soil C stocks were significantly different. Generalised linear model (GLM) was fitted to test differences in soil C content in soil fractions (POM >53 µm; HF >53 µm and S+C <53 µm) in response to management changes (NV-to-V and B-to-UB) for 0–10; 10–20 and top 0–20 cm depth. All statistical analysis was carried out in the R programming language 3.4.3 (*R Development Core Team, 2018*).

For the CENTURY modelling approaches the correlation coefficient (r) and root mean square error (RMSE) were used to compare simulated and measured soil C stocks, in accordance with *Smith et al. (1997)*.

# RESULTS

## Soil C stock

### No vinasse (NV) to vinasse (V)

The differences in soil C stocks between vinasse management practices were significant at shallow depths in the soil profile (Fig. 2). The V management site had significantly more soil C stocks at all points in the profile from 0 to 30 cm depth whereas the NV had slightly more soil C from 60 to 80 cm below the soil surface (Fig. 2). The other layers (40–50, 80–90 and 90–100 cm) did not differ between NV and V site. Within the top 100 cm, the highest soil C stocks were found in the top 10 cm, with 31.0 and 22.6 Mg C ha$^{-1}$ under V and NV, respectively.

These differences in specific depths also resulted in a response ratio higher than 1 for the three-layer increments 0–30, 0–50 and 0–100 cm (1.10; 1.14 and 1.09 for, respectively) indicating soil C stock gain after a change in the management practice (Table 1). Based on vinasse being first applied early in 2004, the management practice change from NV to V led to an accumulation of soil C of 0.55, 1.11 and 1.09 Mg C ha$^{-1}$ yr$^{-1}$, for 0–30, 0–50 and 0–100 cm, respectively. These increments represent C sequestration of 2.0, 4.1 and 4.0 Mg CO$_2$-eq ha$^{-1}$ yr$^{-1}$ between 2004 and 2013 and a decrease in the payback time for soil C stocks in ethanol production under vinasse-based management for the 0–30, 0–50 and 0–100 cm layer intervals (Table 2).

### Burned (B) to Unburned (UB)

Soil C stocks increased significantly following cessation of burning managements from 30–60 cm in the soil profile, but did not differ for the top 20 cm of the profile or between 60 and 100 cm (Fig. 3). Under B management, soil C stocks were largest in the surface from

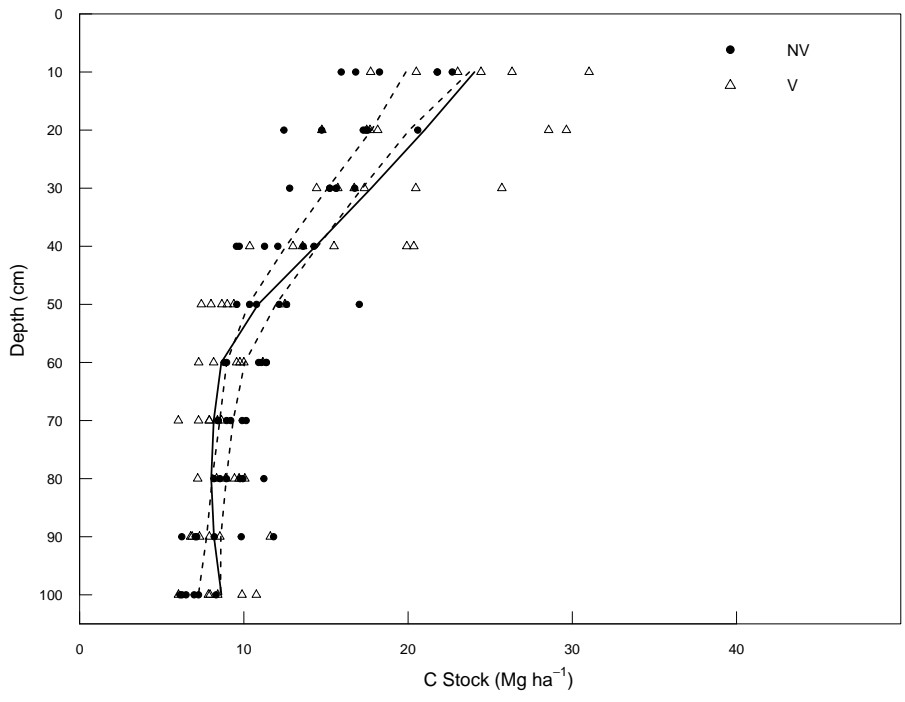

**Figure 2** **Soil C stock (Mg ha$^{-1}$) between 0–100 cm depth in no vinasse and vinasse land management systems.** No vinasse (NV; filled circles); vinasse (V; empty triangles). Dashed lines represent upper and lower bounds of 95% confidence intervals from bootstrapped ($n = 1{,}000$) loess regressions of combined NV and V data; solid lines represent loess regression of soil C stocks in V only, where the line sits outside the confidence interval it can be inferred that NV and V are significantly different.

**Table 1** **Effect of management change on soil carbon stocks (Mg ha$^{-1}$), management change factor and C sequestration potential in the depth increments 0–30, 0–50 and 0–100 cm no vinasse (NV), vinasse (V), burned (B), unburned (UB).** Standard error for 0–30 cm depth increment based on $n = 12$, while for 0–50 and 0–100 cm depth based on $n = 6$.

| Management change comparison | Time span (years) | Depth (cm) | Soil C stocks (Mg ha$^{-1}$) | | Management change factors | C sequestration (Mg CO$_2$ ha$^{-1}$) | C sequestration rate (Mg CO$_2$ ha$^{-1}$ y$^{-1}$) |
|---|---|---|---|---|---|---|---|
| | | | **Previous management** | **New management** | | | |
| NV to V | 10 | 0–30 | $54.8 \pm 0.3$ | $60.3 \pm 0.7$ | 1.10 | 20.0 | 2.0 |
| | | 0–50 | $75.2 \pm 0.9$ | $86.3 \pm 1.7$ | 1.14 | 40.6 | 4.1 |
| | | 0–100 | $119.5 \pm 1.0$ | $130.4 \pm 1.8$ | 1.09 | 40.2 | 4.0 |
| B to UB | 12 | 0–30 | $63.1 \pm 0.6$ | $65.2 \pm 0.4$ | 1.03 | 7.8 | 0.7 |
| | | 0–50 | $81.8 \pm 1.4$ | $91.8 \pm 0.9$ | 1.12 | 33.1 | 2.8 |
| | | 0–100 | $117.0 \pm 1.5$ | $121.7 \pm 1.1$ | 1.04 | 17.3 | 1.4 |

0–10 cm (28.2 Mg C ha$^{-1}$). Taking into consideration that management change from B to UB occurred in 2001, 12 years before soil sampling, it led to an increase of ~0.7 Mg C ha$^{-1}$ yr$^{-1}$ for the depth interval from 30 to 60 cm depth.

This soil C stock increase at specific depths led to management change response ratios greater than 1 for all of the layer intervals assessed after management change (1.03; 1.12

**Table 2 Adjusted payback time for sugarcane ethanol considering both land use conversions assessed by *Mello et al. (2014)* and the effects of management change potential.** Depth increments 0–30, 0–50 and 0–100 cm no vinasse to vinasse (NV-to-V sites, respectively), burned to unburned (B-to-UB sites, respectively). NC means not calculated by *Mello et al., 2014*.

| Land use conversion | Depth (cm) | C debt (Mg $CO_2$ $ha^{-1}$) in 20 years (*Mello et al., 2014*) | Average payback time (years) (*Mello et al., 2014*) | Recalculate payback time (years)[a] | |
|---|---|---|---|---|---|
| | | | | NV-vs-V | B-vs-UB |
| | 0–30 | 20.7 | 2.1 | 1.8 | 2.0 |
| Pasture | 0–50 | 26.8 | 2.7 | 1.9 | 2.1 |
| | 0–100 | 31.8 | 3.2 | 2.3 | 2.8 |
| | 0–30 | 77.2 | 7.9 | 6.5 | 7.4 |
| Cerrado | 0–50 | NC | NC | NC | NC |
| | 0–100 | NC | NC | NC | NC |

**Notes.**
[a]Taking into account the total C debt found by *Mello et al. (2014)* due to LUC after 20 years and our findings regarding the management change plus sugarcane ethanol offset of 9.8 Mg CO2 $ha^{-1}$, $yr^{-1}$ reported by *Fargione et al. (2008)*.

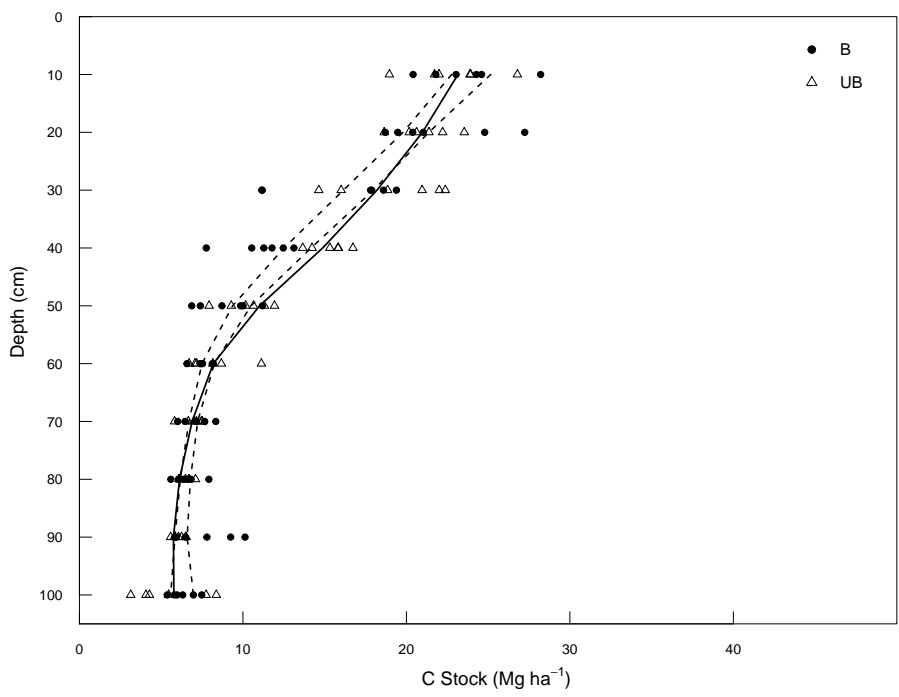

**Figure 3 Soil C stock (Mg $ha^{-1}$) between 0–100 cm depth in Burned and Unburned land management systems.** Burned (B; filled circles); unburned (UB; empty triangles). Dashed lines represent upper and lower bounds of 95% confidence intervals from bootstrapped ($n = 1,000$) loess regressions of combined B and UB data; solid lines represents loess regression of soil C stocks in UB only, where the line sits outside the confidence interval it can be inferred that B and UB are significantly different.

and 1.04 for 0–30, 0–50 and 0–100 cm, respectively) (Table 1). This result also suggests C accumulation of 0.17, 0.83 and 0.40 Mg C ha$^{-1}$ yr$^{-1}$ and a sequestration rate of 0.7, 2.8 and 1.4 Mg CO$_2$-eq ha$^{-1}$ yr$^{-1}$ between 2002 and 2013 for the 0–30, 0–50 and 0–100 cm layer intervals, respectively. Consequently, this also resulted in a decrease in the time required for the sugarcane plantation to achieve a net C benefit under unburned management (Table 2).

## Carbon in soil fractions
### No vinasse (NV) to vinasse management (V)
The average mass balance recovery of physical fractionation ranged between 97% and 98% (Table S6). Higher soil C concentration was found in the S + C than either of the other >53 µm fractions for both NV and V management for the 0–10 and 10–20 cm layers and consequently for the top 0–20 cm depth (Fig. 4). An enhanced C concentration was observed for 0–10 and 10–20 cm depth under V management, which results in an increase of roughly 20% in S + C fraction for the top 20 cm depth, however, it did not significantly differ from NV management by the GLM ($p > 0.05$). The same was observed in the POM fraction, where the V management resulted in higher soil C concentration at 10–20 cm depth but not statistically verified ($p > 0.05$). For fraction and depths where NV had higher soil C concentration, POM 0–10 and HF 10–20 cm depth, again the results were not statistically significant at either case by the GLM ($p > 0.05$) (Fig. 4).

### Burned (B) to unburned (UB)
Similar to the observations of vinasse-based management, a higher soil C concentration was found in the S + C at 0–10 and 10–20 cm depth than either of the >53 µm fractions for both B and UB management practices (Fig. 5). Considering the top 20 cm depth, UB management led to a significant increase C concentration in S + C fraction ($p < 0.05$) (Fig. 5C). Interestingly, the opposite was observed for both POM and HF fractions. The use of B management lead to higher soil C concentration in the 0–10 and 10–20 cm depth as well as to the top 20 cm depth, though these differences were not statistically significant different ($p > 0.05$).

## Modelling soil C
Model simulations showed a good fit between simulated and measured values of soil C stocks (0–20 cm depth) between all sugarcane sites assessed ($r^2$ value $= 0.90$, $n = 4$ and RMSE $= 1.72$) (Fig. 6). Despite the model simulation underestimating empirical yield measurements (Fig. S5), all study sites are within the measured variability and clearly demonstrate that the simulations follow the measured trend for either management comparison.

Measured and modelled soil C stocks of the native forest for top 20 cm depth (53.6 and 52.5 Mg ha$^{-1}$ respectively, Fig. 6) showed that at the time of measurement (2013), this land use had a higher soil C stock than all sugarcane sites, regardless of the management practice used. Simulating LUC from native vegetation to pasture (34 years under pasture) predicted an average soil C loss of $-0.25$ Mg ha$^{-1}$ yr$^{-1}$ (Fig. 7A). The highest soil C losses were observed where an annual crop, mainly soybean, replaced pasture. This LUC followed

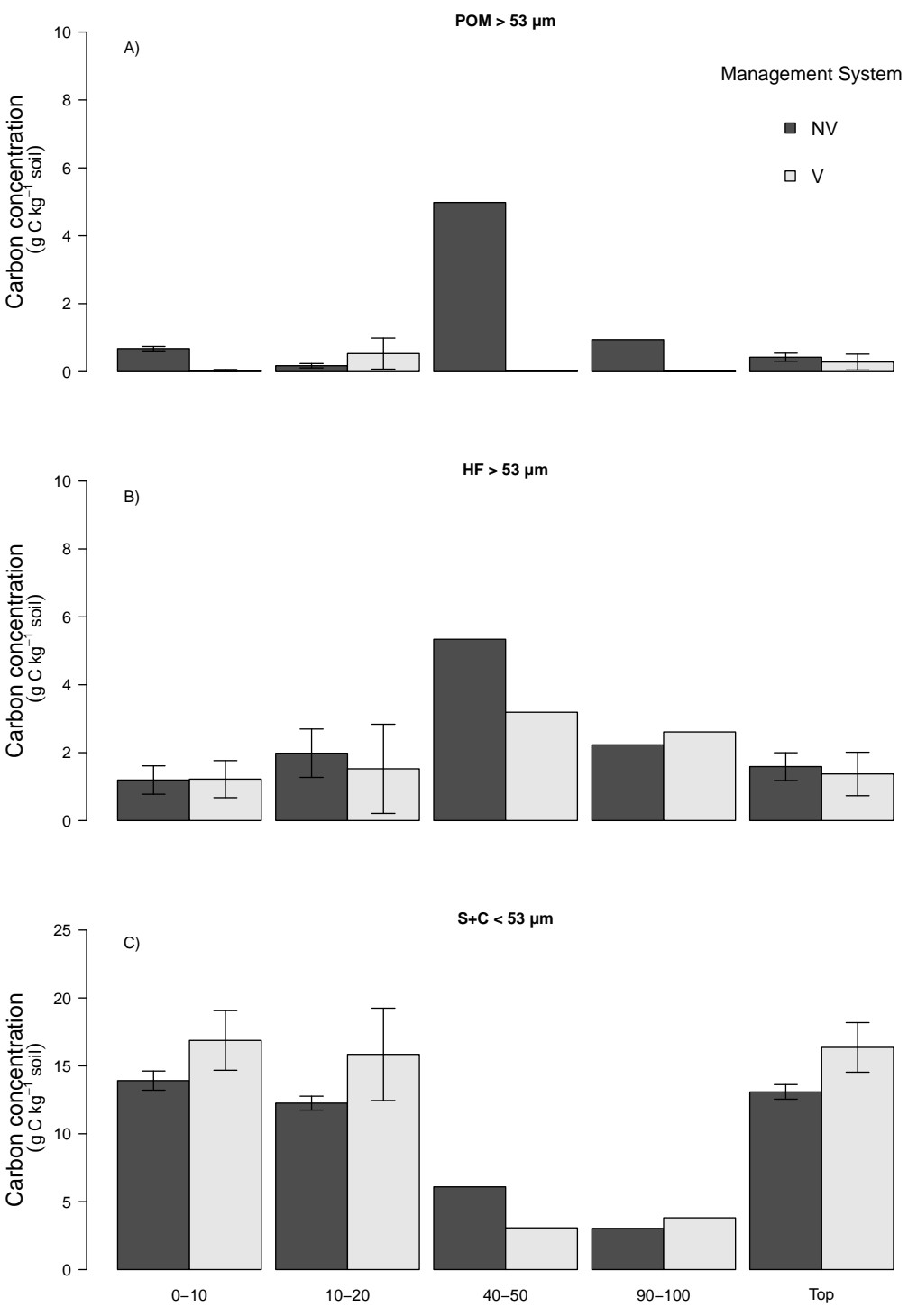

**Figure 4** **Carbon concentration (g C kg⁻¹ soil) in particulate organic matter, heavy and silt + clay soil fractions for no vinasse and vinasse application managements.** (A) Particulate organic matter (POM > 53 μm), (B) heavy (HF > 53 μm), (C) silt + clay (S + C < 53 μm), no vinasse (NV), vinasse (V). Top means 0–20 cm depth. Vertical bars show ±1 standard error ($n = 3$), no vertical bars ($n = 1$).

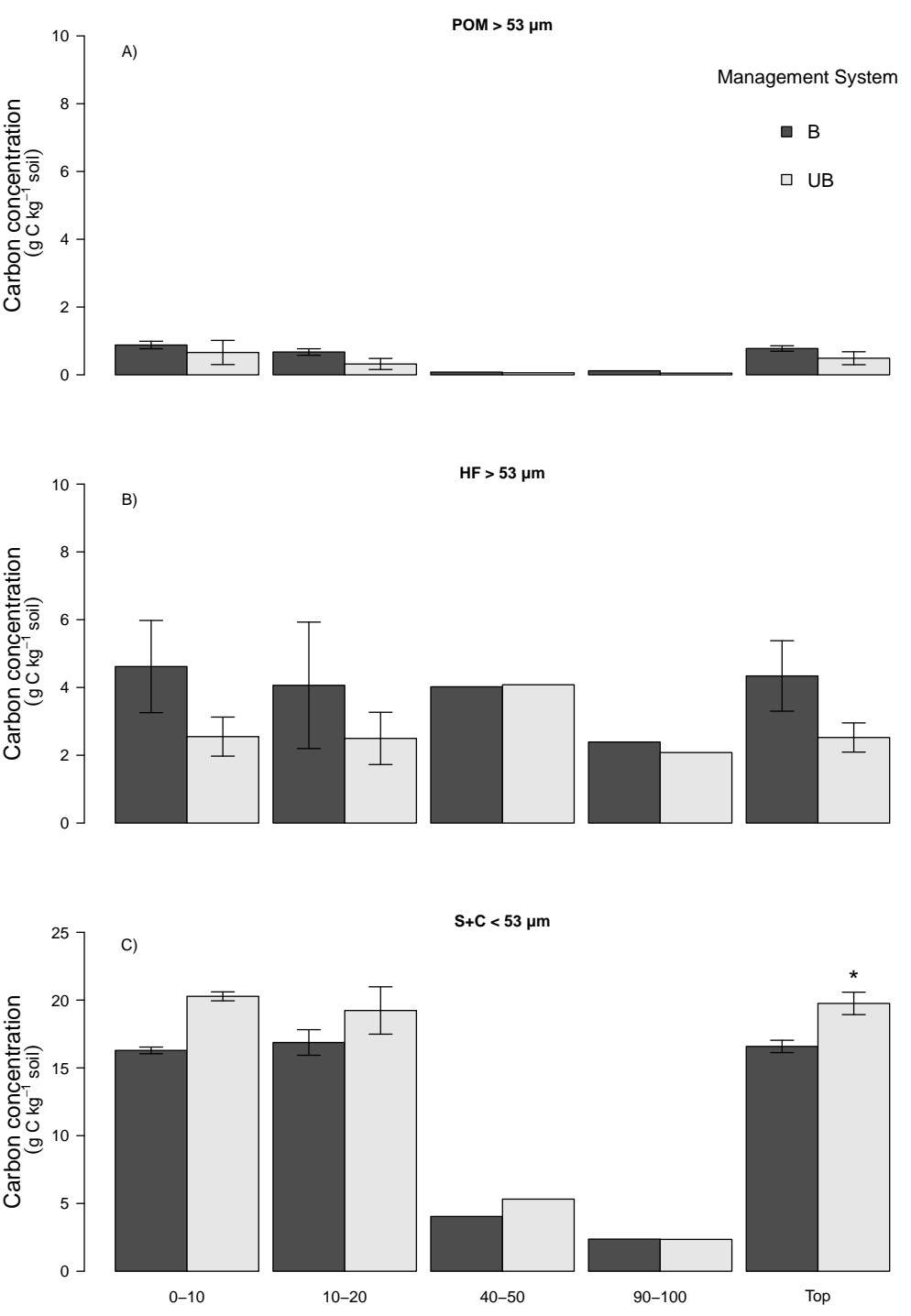

**Figure 5   Carbon concentration (g C kg⁻¹ soil) in the particulate organic matter, heavy and silt + clay soil fractions for burned and unburned management.** (A) Particulate organic matter (POM > 53 $\mu$m), (B) heavy (HF > 53 $\mu$m), (c) silt + clay (S + C < 53 $\mu$m), burned (B), unburned (UB). Top means 0–20 cm depth. Vertical bars show ±1 standard error ($n = 3$), no vertical bars ($n = 1$). Asterisk denote significant difference between land management systems ($P < 0.05$).

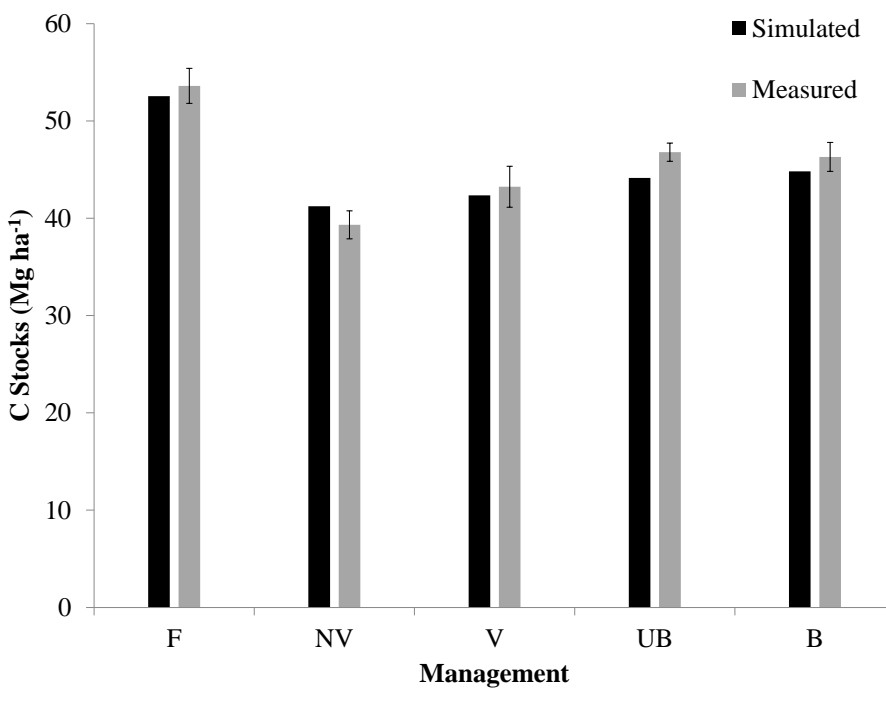

**Figure 6  Simulated and measured results for soil C stock (0–20 cm) under different sugarcane management systems and native vegetation.** No vinasse (NV), vinasse (V), burned (B), unburned (UB), native vegetation (F, as measured by *Franco et al., 2015*). Vertical bars show ±1 standard error ($n = 12$).

by 10 years under soybean resulted an average of soil C loss of $-1.5$ Mg ha$^{-1}$ yr$^{-1}$ (Fig. 7B). Soil C losses were also observed when the burned sugarcane plantations replaced pasture ($-0.25$ Mg ha$^{-1}$ yr$^{-1}$, Figs. 7A, 7B) while the opposite trend was observed when sugarcane replaced annual crops, with soil C gains of 0.35 Mg ha$^{-1}$ yr$^{-1}$ (Fig. 7B).

The measured soil C stock differences between management practices in topsoil (0–20 cm) were also observed in model simulations. In 2013 (year of soil sampling) simulation results showed that vinasse application had higher soil C stocks than the site without vinasse (42.3 and 41.2 Mg ha$^{-1}$ respectively), and little difference between B and UB managements for 0–20 cm depth (44.8 and 44.1 Mg ha$^{-1}$, respectively) (Fig. 6). Since simulated soil C stocks were similar to measured values, it helps to justify the use of an 87-year projection from 2013 to 2100.

Our simulated results indicate that sugarcane with vinasse application leads to an increase of soil C of 0.64 Mg ha$^{-1}$ yr$^{-1}$ from 2013 up to 2035 before a new equilibrium soil C stock is reached; when no vinasse was applied, C stocks increased by 0.52 Mg ha$^{-1}$ yr$^{-1}$ in the same interval period (Fig. 7A). Subsequently, the model predicted minimal differences between the V and NV managements between 2035 and 2100 (3.0–3.5 Mg C ha$^{-1}$ for the top 20 cm) (Fig. 7A). Long-term projections also showed increased sugarcane yields under vinasse application (Fig. S5).

From 2050 onwards, the topsoil C stocks were relatively stable in both B and UB management practices, but the burned site reported 23 Mg C ha$^{-1}$ less than the unburned

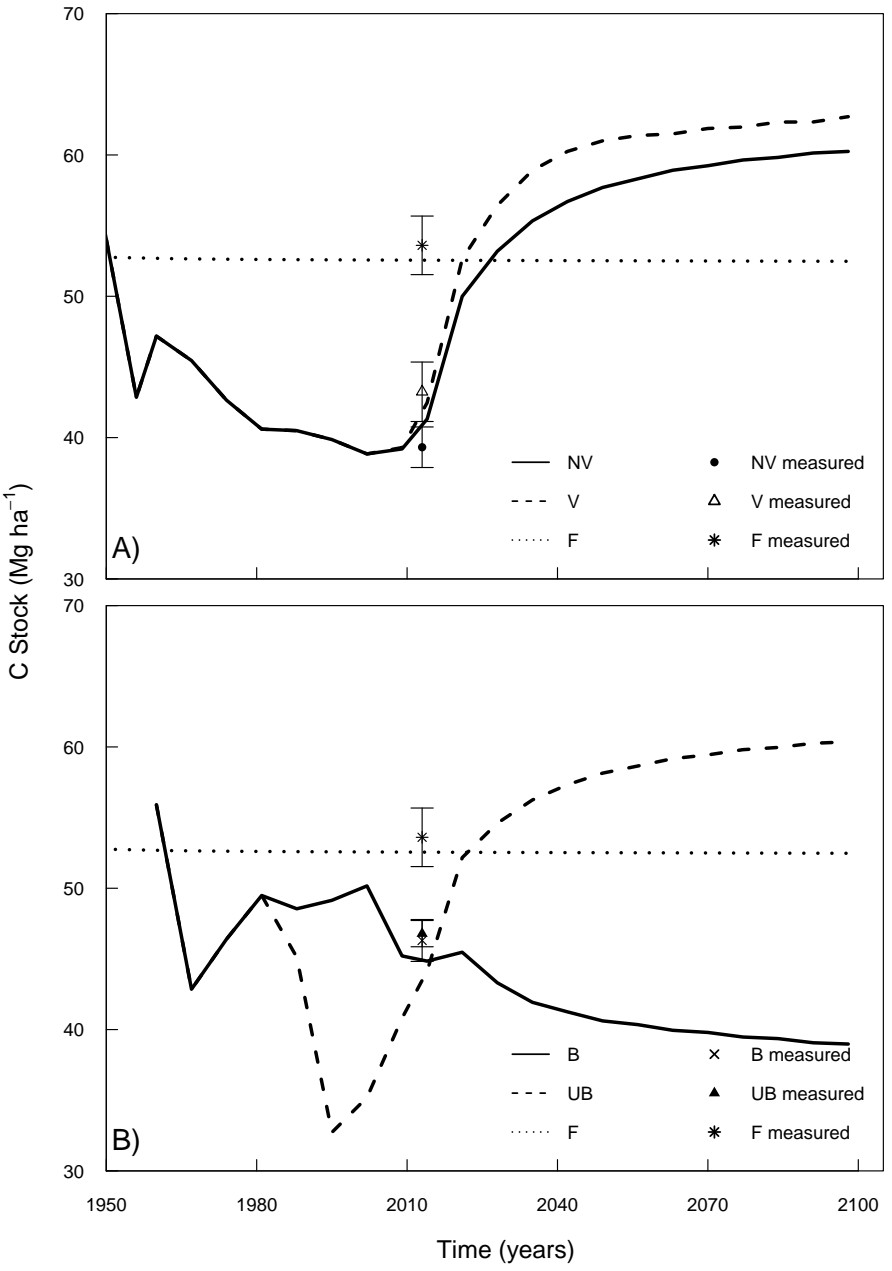

**Figure 7** **Long-term simulations of soil C stocks over 87 years projection in the 0–20 cm soil depth including measured points (2013) for the management change situations.** (A) NV *to* V (management change from no vinasse to vinasse-based management) and (B) B *to* UB (management change from burned to unburned management). F is the native vegetation sampled by *Franco et al. (2015)*. Vertical bars show ±1 standard error (*n* = 12).

site by 2100. The C stock in the unburned site increased by approximately 0.4 Mg ha$^{-1}$ yr$^{-1}$ up to the equilibrium state, from 2013 to 2050 (37 years), while the burned site reached a near-equilibrium state rapidly following a decrease of soil C stock by 0.12 Mg ha$^{-1}$ yr$^{-1}$ for the same period. Although burned management have had higher measured yields

until 2015, a long-term simulation (2015–2100) indicates that unburned management will increase yield potential in relation to burned site (Fig. S5).

Model simulations also indicated that the interaction of vinasse application in burned sugarcane systems may prevent further decreases of soil C stocks (Fig. 8A). There was an increase of approximately 2 Mg C ha$^{-1}$, equivalent to 10%, under lower application rates of vinasse in comparison to a burned management without vinasse additions. Under high vinasse application rates, the increase of soil C may be double (average of 0.04 Mg C ha$^{-1}$ yr$^{-1}$), when compared with the B management, resulting a difference of approximately 5 Mg C ha$^{-1}$ by 2100. The use of different application rates of vinasse in the UB management (Fig. 8B) had a similar pattern observed under a B management. Nevertheless, under UB management, similar soil C stocks were simulated for 2100 whether receiving low or standard application rates.

## DISCUSSION

### Effects of vinasse application on soil C

Our results suggest that vinasse application can increase soil C stocks, particularly in the top 30 cm. This increase may be partly due to enhanced sugarcane biomass production (*De Resende et al., 2006*) leading to increased organic matter inputs (*Araújo et al., 2009*; *Brandani et al., 2015*) and nutrients such as exchangeable K (*Zolin et al., 2011*). Increased biomass production leads to increased plant litter input which in turn can modify soil quality through improved physical, chemical and biological properties (*Jiang et al., 2012*; *Prado, Caione & Campos, 2013*) resulting in more soil C being accumulated (*Galdos, Cerri & Cerri, 2009*). Yield improvement following vinasse application have been observed to be approximately 13% greater in the first and second cycles of sugarcane plantation (*De Resende et al., 2006*).

The infiltration of vinasse soluble C and nutrients is likely to be limited to the soil surface layers where more than 80% of the root system can dominate (*Ohashi et al., 2015*). In these surface layers the greater availability of nutrients, organic matter and water provided by vinasse may stimulate higher root biomass density by reducing water and nutrient limitation (*Smith, Inman-Bamber & Thorburn, 2005*). The results of *Pina et al. (2015)* support this assumption with enhanced root growth with vinasse application compared to basic fertilization (only NKP mineral fertiliser) at 0–20 cm depth. *Jiang et al. (2012)* report that after three years of vinasse application soil porosity was maintained or even increased with greater aggregation of fine soil particles at 0–30 cm depth, which would allow higher potential root growth. Overall an increased concentration of roots in the topsoil layers after vinasse application might stimulate the accumulation of soil C through greater root turnover and exudation as well as enhancement of microbial biomass (*Navarrete et al., 2015*; *Pina et al., 2015*). By contrast, lower nutrient availability in the surface layers under no vinasse management may encourage root development in deeper soil resulting in a higher soil C stocks for 60–80 cm depth.

Whilst vinasse application management resulted in higher soil C stocks, this did not translate into a significant soil C accumulation in specific soil fractions as expected. This

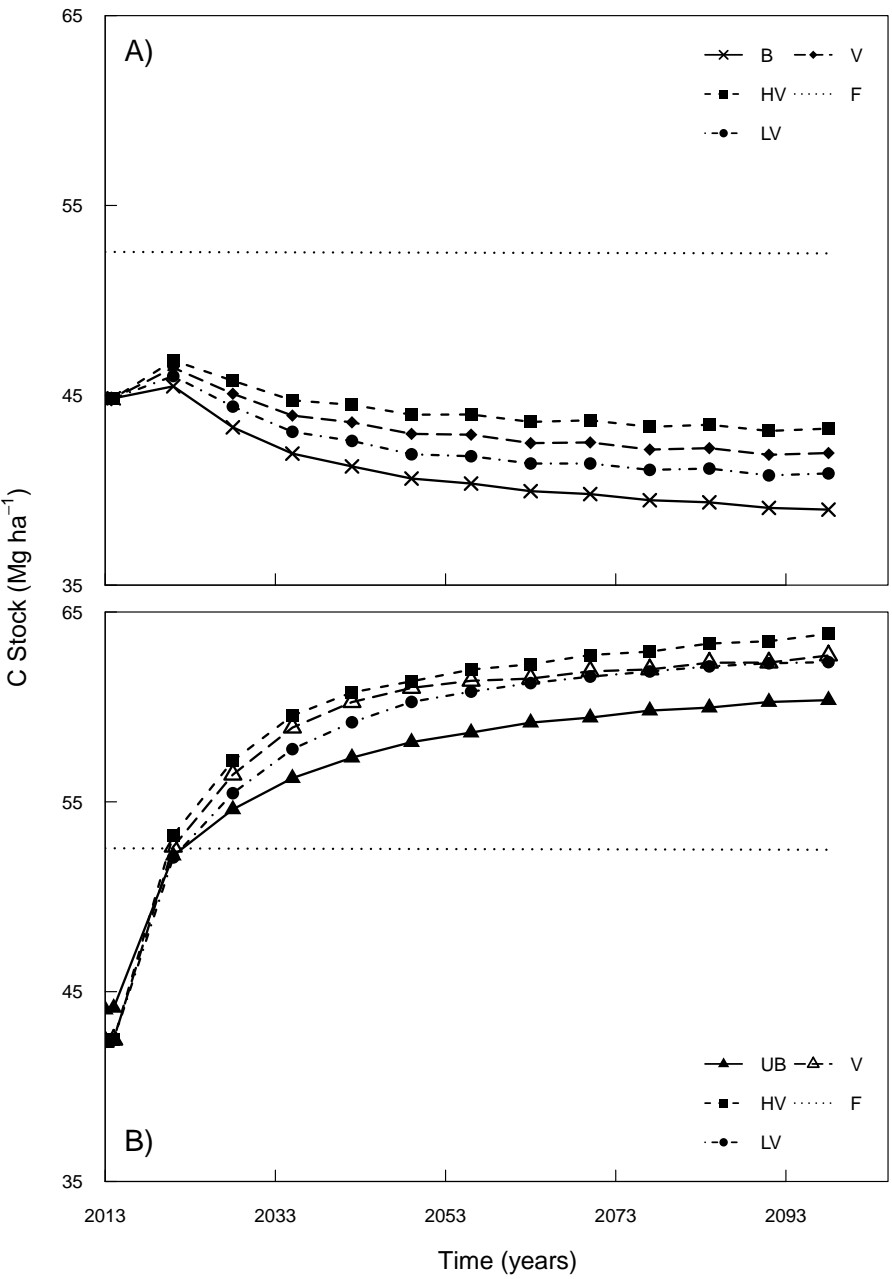

**Figure 8** **Long-term simulations of soil C stocks (0–20 cm) over 87 years projection for burned system and unburned system with different dosages of vinasse.** (A) burned system (B), unburned system B refers to the burned sugarcane without vinasse, UB refers to the unburned sugarcane without vinasse. High vinasse dosages (HV), low vinasse dosages (LV), standard vinasse dosages (V). F is the native vegetation from *Franco et al. (2015)*.

might be associated with the labile nature of the C that vinasse added and potential fast decomposition of it. Greater nutrient availability promoted by vinasse management can increase the soil decomposers community which may easily offset the C added (*Carmo et al., 2013*; *Oliveira et al., 2013*; *Siqueira Neto et al., 2016*). Further, the light-coarse particulate organic matter fraction (i.e., POM >53 µm), which is chiefly related to inputs from biomass and plant litter, can also show high spatial and seasonal variability (*Christensen, 1992*). Hence, POM may be seen to represent faster C turnover and nutrient cycling in the soil (*Gama-Rodrigues & Gama-Rodrigues, 2008*) due to its lability, allowing for rapid microbial decomposition. The lack of difference between no vinasse and vinasse-based management in the fractions (specially POM and HF) might be also associated with the recent soil tillage and replanting operation at both sites and potential exposure of the C added to mineralisation (*Paustian, Collins & Paul, 1997*; *Six et al., 2002*).Therefore, although not significant in this study, the trend in soil C accumulation at S+C fraction under vinasse management is an important observation being a potentially stable soil fraction. (*Christensen, 2001*; *Lisboa et al., 2009*).

In general, vinasse application was a beneficial management practice. Its application led to an accumulation of 0.55 Mg ha$^{-1}$ yr$^{-1}$ soil C stock of over 10 years in the topsoil (0–30 cm) layers. This is more than double of the soil C accumulation found by *De Resende et al. (2006)* in a related study over 16 years for the top 20 cm of soil (0.25 Mg ha$^{-1}$ yr$^{-1}$). Whilst an accumulation of soil C of 1.1 Mg ha$^{-1}$ yr$^{-1}$ was found at 1 metre depth along, there are still uncertainties about the effect of vinasse beneath the topsoil. This overall increase in soil C stocks is an important part of C accounting in managed sugarcane plantations, and therefore is important to be considered in any future life-cycle analyses and calculation of payback time.

## The role of burning cessation on soil C

Cessation of burning (UB management) for a 12 year period from 2001 showed higher soil C stock at 30–60 cm depth but no difference for 0–20 and 70–100 cm depth. Our findings contrast other studies which report significant differences between burned and unburned management in the topsoil layers only (0–30 cm) (*Cerri et al., 2011*; *Galdos, Cerri & Cerri, 2009*; *Signor et al., 2014*). The legacy effects of past land use change and management at this particular study site may have influenced the observed C stocks within the soil profile.

At the UB management part of the study site, annual cropping was used for 10 years prior to sugarcane plantation. At the other part of the study site where B management was still in place the prior land was under pasture/grassland during the same period. According to *Zinn, Lal & Resck (2005)* and *Maia et al. (2010)*, cropland can experience consistently higher losses of C due to continued disturbance, while pasture may either decrease or increase soil C storage relative to native vegetation areas (*Maia et al., 2009*). Additionally, in this study sugarcane replanting operation occurred in different times between the sites (2011 and 2008 for UB and B, respectively) and may have homogenised the topsoil layers (0–20 cm). The replanting operation includes land preparation every 6 years by plowing, disking and in some cases subsoiling prior to planting and fertiliser applications. During replanting, the organic matter in aggregates are exposed due to physical disruption,

facilitating increased soil C mineralisation (*Paustian, Collins & Paul, 1997*; *Six et al., 2002*), which may result in up to 80% C losses in the first 0–20 cm depth (*Silva-Olaya et al., 2013*). Soil tillage was suggested as factor for the lack of significant differences found in soil C (*Galdos, Cerri & Cerri, 2009*) or only a modest accumulation rate (*Cerri et al., 2011*) in topsoil layers. Higher C content may also be found in topsoil under burned management due to the presence of charcoal and ash on the soil surface (*Blair, 2000*). Tillage operations, nonetheless, may have been a primary cause for the higher soil C stocks recorded in the UB site from 30 to 60 cm depth through incorporation of the straw deposited in the surface of the soil. According to *Thorburn et al. (2011)* the amount of straw deposited on the soil surface in the unburned management can reach about 10 to 20 Mg ha$^{-1}$ with soil under this management often being of higher fertility than soil under burned management (*Correia & Alleoni, 2011*). Accelerated rates of root turnover due to preferable conditions created by a deeper litter layer (e.g., humidity, temperature and the presence of microorganisms) is expected under unburned management and may help to rationalise the increase in the soil C stock between 30 and 60 cm. Improved aggregate stability and/or potential additions of other composts to the planting groove (usually done around 40–50 cm depth) may also lead to higher soil C stocks at these depths. In this study, the in deep soil layers (30–60 cm) after management change from B to UB suggests an increase of soil C stock of approximately 0.7 Mg ha$^{-1}$ yr$^{-1}$. This results supports the work of others which suggest a deeper soil layers need to be considered when making conclusions around land use and management impacts on soil C stocks (*Keith et al., 2015*; *Mello et al., 2014*).

The recent soil tillage might also have influenced the distribution of the soil C between fractions. *Six, Elliott & Paustian (1999)* emphasised that soil disturbance might alter the aggregate dynamics, increasing the turnover time of SOM and, thus, decreasing the formation of stabilised C fractions. *Signor et al. (2014)* used the same physical fractionation method as in this study, and showed lower C concentration in the POM under sugarcane with recent soil disturbance, whereas areas under less disturbance had higher C concentration, regardless of whether the sugarcane was burned or unburned. Our results agree with these findings, indicating a somewhat higher C content in the POM and HF for B compared to UB management where the tillage occurred recently, although they do not differ statistically. The POM is regarded as the main driver responsible for the stabilisation of the macro-aggregates in the soil and has an important role in determining changes of the total C (*Tisdall & Oades, 1982*). Thus, we suggest that the statistical differences found in S + C in topsoil (0–20 cm) under the UB management occurred due to decomposition processes at POM prior to soil disturbance i.e., tillage which captured part of this fraction at the S + C fraction (*Roscoe et al., 2001*).

An increase of SOM and consequently in soil C stocks, or at least the maintenance of their levels, would lead to higher yields (*Pan, Smith & Pan, 2009*). Therefore, Brazilian sugarcane under an improved UB management could positively contribute to issues about food security, energy demands and climate changes (*Goldemberg et al., 2014*; *Lal, 2004*).

## Long-term scenarios of improved sugarcane management

The modelling predictions support empirical data suggesting that improved practices could increase soil C stocks in the topsoil (20 cm) over a longer-term. Simulated outcomes (0–20 cm) for prior land uses and LUC for all sites assessed are similar to other work. For example, *Maia et al. (2009)* found a similar rate of soil C loss for 0–30 cm depth, in degraded pastures ($-0.28$ Mg ha$^{-1}$ yr$^{-1}$) previously covered by Amazon forest and Cerrado vegetation. *Silva-Olaya et al. (2016)* in a simulated study with CENTURY, reported that LUC from pasture to sugarcane resulted in losses of soil C of $-0.19$ Mg ha$^{-1}$ yr$^{-1}$ (0–20 cm). For LUC from pasture to cropland (principally soybeans), the annual decrease of soil C losses found in our simulation is also in aligns with measured results found by *Carvalho et al. (2010)* and simulated outcomes of *Silva-Olaya et al. (2016)*.

After the introduction of sugarcane and the improved practices, an increase in soil C stock was noted in the long-term simulations (0–20 cm). Transition from NV to V management indicated an upward trend over 22 years (2013–2035) of simulation. It is notable that soils under V management showed a higher potential for C stocks to increase, albeit at a lower rate as the soils approached an equilibrium C level. Similar measured results were detected in topsoil layers (0–20 cm) by *De Resende et al. (2006)*, studying 16 years of vinasse plus unburned management, with the authors reporting an increase soil C stock of around 4 Mg ha$^{-1}$ over this period. *Brandani et al. (2015)* also simulated similar outcomes with CENTURY for unburned with additional organic amendments including vinasse. Ultimately, the difference of only 3–3.5 Mg ha$^{-1}$ between NV and V sites found after 22 years of simulation, indicated that soil C stocks under vinasse-based management will depend on the balance between increased C inputs and outputs related to potential change in microbial activity, as well as whether the site is under burned or unburned management. Our results indicate that vinasse application under burned management could be to offset a burning associated decline in soil C stocks (Fig. 8A). Soil C stock was 30% lower than native vegetation under a burned management, but when combined with vinasse application the difference was only 20% at high application rates. While vinasse application may have some benefits to increase soil C stocks, other issues need to be considered such as salinisation and GHG emission (*Christofoletti et al., 2013*; *Oliveira et al., 2013*).

Long-term simulations showed that unburned management has great potential to increase topsoil C stocks (0–20 cm) as well as sugarcane yield (Fig. 7 and Fig. S5, respectively). Over a projected 35 year period (2013–2049), the increase of soil C stock from B to UB management was approximately 0.4 Mg ha$^{-1}$ yr$^{-1}$, followed by a slower rate of accumulation ($\sim$0.06 Mg ha$^{-1}$ yr$^{-1}$) as the equilibrium state for soil C storage was achieved (2050–2100). These results agree with *Razafimbelo et al. (2006)* who measured a C accumulation rate of 0.65 Mg ha$^{-1}$ yr$^{-1}$ in the topsoil layer (10 cm) for an unburned area after 6 years (one cycle without replanting operations, ploughing and disking); the authors suggested further research is required to assess the effects of tillage disturbance on replanting sugarcane. Our simulated long-term results indicated that the increase in C accumulation may be the same for 0–20 cm depth even taking into consideration tillage every 6 years, during a period of 35 years, i.e., six tillage operations carried out. All simulated long-term perspectives should be considered as scenarios only since as uncertainties around

future crop varieties, climatic variation and management options will all impact on the future performance of these bioenergy systems.

## CONCLUSIONS

In this study we examined the impacts of vinasse application and the cessation of burning on soil C, demonstrating both empirically and with a mechanistic model, potential benefits to soil C stocks after change in sugarcane management practices. Vinasse application might be the most available first step to mitigate losses of soil C stocks in burned sugarcane plantations. Although this study provides a site specific approach, the outcomes build on the payback time calculation for ethanol production by *Mello et al. (2014)*. This is particularly relevant for the State of São Paulo being responsible for approximately 60% of Brazil's production and where such management changes are being widely adopted (*UNICA, 2016*). However, further studies are highly required, particularly considering Brazil's central region where the sugarcane expansion is widely occurring (*Adami et al., 2012*) as well as taking into account other soil types, crop varieties, management practices and climate conditions.

## ACKNOWLEDGEMENTS

The authors thank São Luiz mill to provide the areas for the study, and also Adriana M. Silva-Olaya for the suggestions in the modelling approaches and Peter A. Henrys (CEH) for useful discussions and input on statistical approaches.

### Funding

This study was mainly supported by the São Paulo Research Foundation (FAPESP) and the National Council for Scientific and Technological Development (CNPq) Brazil (projects 2013/12600-2 and 2013/24506-0). Staff from the Centre for Ecology & Hydrology (CEH) were supported by CEH National Capability projects NEC05570, NEC05593 and the EPSRC funded MAGLUE project (http://www.maglue.ac.uk). There was no additional external funding received for this study. The funders had no role in study design, data collection and analysis, decision to publish, or preparation of the manuscript.

### Grant Disclosures

The following grant information was disclosed by the authors:
São Paulo Research Foundation (FAPESP).
National Council for Scientific and Technological Development (CNPq) Brazil: 2013/12600-2, 2013/24506-0.
CEH National Capability projects: NEC05570, NEC05593.
EPSRC funded MAGLUE project.

### Competing Interests

The authors declare there are no competing interests.
## Author Contributions

- Caio F. Zani conceived and designed the experiments, performed the experiments, analyzed the data, contributed reagents/materials/analysis tools, prepared figures and/or tables, authored or reviewed drafts of the paper, approved the final draft.
- Arlete S. Barneze conceived and designed the experiments, performed the experiments, analyzed the data, authored or reviewed drafts of the paper, approved the final draft.
- Andy D. Robertson and Aidan M. Keith analyzed the data, authored or reviewed drafts of the paper, approved the final draft.
- Carlos E.P. Cerri and Carlos C. Cerri conceived and designed the experiments, analyzed the data, contributed reagents/materials/analysis tools, authored or reviewed drafts of the paper, approved the final draft.
- Niall P. McNamara analyzed the data, contributed reagents/materials/analysis tools, authored or reviewed drafts of the paper, approved the final draft.

## Data Availability

The raw data are provided in Supplemental Information 1.

## Supplemental Information

Supplemental information for this article can be found online at http://dx.doi.org/10.7717/peerj.5398#supplemental-information.

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
