# Peer review of "Vinasse application and cessation of burning in sugarcane management can have positive impact on soil carbon stocks"

_PeerJ, doi:10.7717/peerj.5398_

## Round 0.1 · original submission · Major Revisions

You will see that one reviewer recommended minor revisions while the other one recommended me to reject the manuscript. I think they both have valid arguments so I would like to give you the chance to prepare a new version of the manuscript. In particular, both reviewers describe serious issues with the results discussion regarding carbon storage. Please clarify this point in the next version.

Reviewer 1 ·

Basic reporting

no comment

Experimental design

The experimental design is good. The approach, sampling method and measurements are well designed. The description for the method is very detailed and understandable. Author needs to supplement the detailed information of UB treatment for straw deposition as stated in my attachment.

Validity of the findings

no comment

Additional comments

The manuscript is well designed and is well written in the part of Introduction and Methods and Materials. However, I think there are some serious problems.
1. The conclusion of Vinasse application and cessation of burning in sugarcane management having positive impact on soil carbon stocks is questionable. Although Fig 2 and Fig 3 shows that some individual soil horizons have a C accumulation after vinasse application or cessation of burning, Table 1 clearly shows that the difference in soil C stocks between NV-to-V management or between B and UB were not deemed statistically significant for not only top soil but also whole profile of soil (0-30, 0-50 and 0-100 cm). Moreover, Fig 4 shows all of soil fractions (including labile and recalcitrant pool ) in all soil horizon have not significant difference between V and NV treatments. Similarly, Fig 5 shows all of soil fractions in all soil horizons have not significant difference between B and UB treatments except for silt and clay fractions in 0-20 cm horizon. Therefore, based on your results, vinasse application and cessation of burning have a weak impact on soil carbon stocks. Thus, I think it is no meaning to calculate the C sequestration rate or payback time.
2. The authors used the CENTURY model simulate future C change under vinasse application and cessation of burning, and the results showed that C accumulates continually in the future. I am not familiar with modelling, and I cannot judge whether it is reliable.
3. The Discussion part is badly organized and tedious. It did not explain well for the results of this study. Moreover, it did not discuss the limitation of this study (e.g. spatial heterogeneity).
4. The detailed information can be seen in my attachment.

Annotated reviews are not available for download in order to protect the identity of reviewers who chose to remain anonymous.

·

Basic reporting

The research question is significant and important. The data are novel and interesting.

Experimental design

I think the introduction should pay more attention to the impact of different management practices on SOC stock rather than other unrelated content. Besides, author should summarize a better research motivation of this study. And I want to know why not set a control (Native vegetation treatment), this helps to better explain the results.

Validity of the findings

no comment

Additional comments

L21-22 “ mainly regarding soil aspects”, “which requires a better understanding of its effects in a short and long term.” These statements are too broad, the theme of soil carbon should be highlighted, rephrase.

L27 According to your result (L319-320) and discussion (L462), “0-40” should be “0-30”?

L34 “first data?” please confirm

L58-73 L82-92 these two paragraphs are too long and useless, perhaps you should reduce the description about “the vinasse application and burning management situation” and merge this paragraph with the next paragraph, respectively

L69-71 “An appropriate vinasse application, however, requires chemical analyses of the vinasse nutrients, particularly the soil exchangeable K concentration (CETESB 2006).” This sentence is redundant in the introduction. Delete or rephrase.

L81 reword this sentence “It is estimated that over 80% of São Paulo State is under vinasse ferti-irrigation”. Because I do not know the purpose of this sentence appeared at the end of the paragraph

L90-92 delete or reword this sentence “ Despite the cessation burning in São Paulo this practice is still evident in other regions such as in the Brazilian Northeast (States of Pernambuco and Paraíba) and in parts of the State of Minas Gerais state where the uneven topography is present” It is redundant

L214 why not 3 replicates for 90-100 cm?

L321 “;” should be “,” I think

L429-432 There is no need to discuss yield. Besides, vinasse addition stimulates the growth of microorganisms may be another reason that increase C stock, please consider

L434 delete “(”

L442-443 “ the accumulation of soil C through greater root turnover and exudation.” It requires some references

L447 “Silt+Clay containing greater soil C under vinasse application” or “Silt+Clay containing greater soil C under vinasse application and no vinasse application”, check

L447-448 The logic is ambiguous for me. On the one hand, vinasse application significantly increased soil C stocks in 0-40 cm depth, on the other hand, no differences in C content were found in any of the physical fractions. I really want to know the fate of excess C in V treatment, compared with NV treatment.

L451-453 Can you get the result that “LF can show high spatial and seasonal variability” from your study rather than other study (i.e., Christensen 1992)?

L459-460 More profound discussion is necessary

L485 detailed description.

L485-495 The purpose of this paragraph is to support which of your result?

L499-500 “soil under this management often has higher fertility than soil under burned management”, this sentence is contradictory to your result that no significant difference of C stock was founded between B and UB from 0-30 cm.

L510 The definition of “carbon” must be consistent and rigorous. total C is different from “soil organic C (SOC)”, check

L573-585 Delete sentences that have nothing to do with the conclusion

---

## Round 0.2 · Minor Revisions

Thank you for your corrections, I think they improved significantly the manuscript. I would happily accept this manuscript after you work a bit on the formal aspects (check the typos, improve the figures). A quick check by a native speaker would be beneficial.
Please find below some points that could be improved:
- Abstract: it is a bit strange to start an abstract with "Recent studies" since you cannot cite these studies in the abstract. Please rephrase.
- l115: the sentence needs to be rephrased: what is the link between paired site and long-term perspectives?
- L143: FAO classification is sufficient
- L168: compromised means something else
- L185: use instead of utilized
- L199: were used "for the"
- L350: accumulations*
- L474: change facilitated with led to
- L483: The role of burning cessation
- Figures 2-3: it would be more logical to have the depths on the y axis
- Figures 4 and following: the micro symbol is missing on all the figures
- Figure 6: why not comparing measured in y axis and simulated on y axis?

·

Basic reporting

no comment

Experimental design

no comment

Validity of the findings

no comment

Additional comments

I would be very glad to re-review this paper named "Vinasse application and cessation of burning in sugarcane management can have positive impact on soil carbon stocks ". I think the submission has been greatly improved and is worthy of publication. And it is better if the authors further improve figures more beautiful and standardized before submitting the final version .

---

## Round 0.3 · accepted · Accept

Dear author

Thank you very much for taking into account all our comments. I am glad to inform you that I accepted this version of your manuscript.

#